

**The ACER pollen and charcoal database: a global resource to document vegetation and fire**
**response to abrupt climate changes during the last glacial period**
**ACER Project Members\*:** María Fernanda Sánchez Goñi[1,2], Stéphanie Desprat[1,2], Anne-Laure Daniau[3],
Frank C. Bassinot[4], Josué M. Polanco-Martínez[2,5], Sandy P. Harrison[6,7], Judy R.M. Allen[8], R. Scott
Anderson[9], Hermann Behling[10], Raymonde Bonnefille[11], Francesc Burjachs[12], José S. Carrión[13], Rachid
Cheddadi[14], James S. Clark[15], Nathalie Combourieu-Nebout[16], Colin. J. Courtney Mustaphi[17], Georg H.
Debusk[18], Lydie M. Dupont[19], Jemma M. Finch[20], William J. Fletcher[21], Marco Giardini[22], Catalina
González[23], William D. Gosling[24], Laurie D. Grigg[25], Eric C. Grimm[26], Ryoma Hayashi[27], Karin Helmens[28],
Linda E. Heusser[29], Trevor Hill[20], Geoffrey Hope[30], Brian Huntley[8], Yaeko Igarashi[31], Tomohisa Irino[32],
Bonnie Jacobs[33], Gonzalo Jiménez-Moreno[34], Sayuri Kawai[35], Peter Kershaw[36], Fujio Kumon[37], Ian T.
Lawson[38], Marie-Pierre Ledru[14], Anne-Marie Lézine[39], Ping Mei Liew[40], Donatella Magri[22], Robert
Marchant[17], Vasiliki Margari[41], Francis E. Mayle[42], Merna McKenzie[36], Patrick Moss[43], Stefanie
Müller[44], Ulrich C. Müller[45], Filipa Naughton[46,47], Rewi. M. Newnham[48], Tadamichi Oba[49], Ramón
Pérez-Obiol[50], Roberta Pini[51], Cesare Ravazzi[51], Katy H. Roucoux[38], Stephen M. Rucina[52], Louis Scott[53],
Hikaru Takahara[54], Polichronis C. Tzedakis[41], Dunia H. Urrego[55], Bas van Geel[56], B. Guido Valencia[57],
Marcus J. Vandergoes[58], Annie Vincens[11], Cathy L. Whitlock[59], Debra A. Willard[60], Masanobu
Yamamoto[49].
[1]EPHE, PSL Research University, Laboratoire Paléoclimatologie et Paléoenvironnements Marins, F-
33615 Pessac, France, [2]Univ. Bordeaux, EPOC, UMR 5805, F-33615 Pessac, France, [3]CNRS, Univ.
Bordeaux, EPOC, UMR 5805, F-33615 Pessac, France, [4] Laboratoire des Sciences du Climat et de
l'Environnement, LSCE/IPSL, CEA-CNRS-UVSQ, Université Paris-Saclay, F-91191, Gif-sur-Yvette,
France, , [5]Basque Centre for Climate Change - BC3, 48940 Leioa, Spain,[6]Department of Biological
Sciences, Macquarie University, North Ryde, NSW, Australia, [7]School of Archaeology, Geography and
Environmental Sciences (SAGES), Reading University, Whiteknights, Reading, RG6 6AB, UK, [8]
Department of Biosciences, Durham Univ., South Road, Durham DH1 3LE, UK, [9] Environmental
Programs, School of Earth Sciences & Environmental Sustainability, Northern Arizona University,
Flagstaff, AZ 86011, USA, [10]Department of Palynology and Climate Dynamics, Albrecht-von-Haller
Institute for Plant Sciences, University of Göttingen, Germany, [11]CEREGE (UMR 6635), Aix-Marseille
Université, CNRS, IRD, Collège de France, Europole de l'Arbois, BP80, 13545 Aix-en-Provence, France,
[12]ICREA  Barcelona, Catalonia, Spain, Institut Català de Paleoecologia Humana i Evolució Social,
Campus Sescelades URV, W3, 43007 Tarragona, Spain, [13]Departamento de Biología Vegetal, Facultad



de Biología, Universidad de Murcia, 30100 Murcia, Spain, [14]Institut des Sciences de l'Evolution de Montpellier, UMR 5554 Université Montpellier 2, Bat.22, CC061, Place Eugène Bataillon, 34095 Montpellier Cedex 5, France, [15]Duke Trinity College of Art & Sciences, Durham, NC 27708, USA, [16]UMR 7194 CNRS, Histoire naturelle de l'Homme Préhistorique, Département de Préhistoire, Muséum national d'Histoire naturelle, Paris, France, [17]The York Institute for Tropical Ecosystem Dynamics (KITE), Environment Department, University of York, York, Heslington, YO10 5DD, [18]UK, Department of Zoology, Duke University, Box 90325, Durham, NC 27708–0325 U.S.A., [19]MARUM—Center for Marine Environmental Sciences, University of Bremen, D-28359 Bremen, Germany, [20]Discipline of Geography, School of Agricultural, Earth & Environmental Sciences, University of KwaZulu-Natal, Pietermaritzburg, South Africa, [21]Quaternary Environments and Geoarchaeology, Geography, School of Environment and Development, University of Manchester, Oxford Road, Manchester M13 9PL, UK, [22]Dipartimento di Biologia Ambientale, Sapienza Università di Roma, Italy, [23]Departamento de Ciencias Biológicas, Universidad de los Andes, A.A. 4976 Bogotá, Colombia, [24]Palaeoecology & Landscape Ecology, University of Amsterdam, The Netherlands, [25]Geology and Environmental Science Department, Norwich University, VT 05663, USA, [26]University of Minnesota, Department of Earth Sciences, Minneapolis, Minnesota 55455, USA, [27]Lake Biwa Museum, Oroshimocho1091, Kusatsu 525-0001, Japan, [28]Department of Physical Geography and the Bolin Centre for Climate Research, Stockholm University, Stockholm, Sweden, [29]Lamont-Doherty Earth Observatory of Columbia University, Palisades, NY 10601, USA, [30]Department of Archaeology and Natural History, The Australian National University, Fellows Road, Acton ACT 0200, [31]Institute for Paleoenvironment of Northern Regions, Koyocho 3-7-5, Kitahiroshima 061-1134, Japan, [32]Geological Institute, University of Tokyo, Hongo, Bunkyo-ku, Tokyo 113-0033, Japan, [33]Roy M. Huffington Department of Earth Sciences, Southern Methodist University, Dallas, TX 75275-0395, USA, [34]Departamento de Estratigrafía y Paleontología, Universidad de Granada, 18071, Spain, [35]Institute of Mountain Science, Shinshu University, Asahi 3-1-1, Matsumoto 390-8621, Japan, [36]School of Geography & Environmental Science, Monash University, Clayton, Vic 3168, Australia, [37]Department of Environmental Sciences, Faculty of Science, Shinshu University, Asahi 3-1-1, Matsumoto 390-8621, Japan, [38]Department of Geography and Sustainable Development, University of St Andrews, St Andrews, KY16 9AL, UK, [39]LOCEAN - Laboratoire d'Océanographie et du Climat : Expérimentations et Approches Numériques, UPMC, Paris, France, [40]Department of Geosciences, National Taiwan University, 1, Sec. 4, Roosevelt Rd. Taipei 106, Taiwan, ROC, [41]Environmental Change Research Centre, Department of Geography, University College London, London WC1E 6BT, UK, [42]Centre for Past Climate Change, Department of Geography and Environmental Science, University of Reading, Reading, UK, [43]School of Geography, Planning & Environmental Management, The University of Queensland, St Lucia, Australia, [44]Freie Universität Berlin, Geological Sciences, Palaeontology Section, Berlin, Germany, [45]Biodiversity and Climate Research Centre, Senckenberganlage 25, 60325 Frankfurt, Germany, [46]Center of Marine Sciences (CCMAR), Algarve University, Campus de Gambelas, 8005-139 Faro, Portugal, [47]Portuguese Sea and Atmosphere Institute (IPMA), Rua Alfredo Magalh~aes Ramalho 6, 1495-006 Lisboa, Portugal, [48]School of Geography, Environment and Earth Sciences, Victoria University of Wellington, PO Box 600, Wellington 6140, New Zealand, [49]Graduate School of Environmental Earth Science, Hokkaido University, N10-W5 Kita-ku, Sapporo 060-0810, Japan, [50]Unitat de Botànica, Facultat de Biociències, Universitat Autònoma de Barcelona, 08193 Bellaterra, Cerdanyola del Vallès, Spain, [51]C.N.R. – Istituto per la Dinamica dei Processi Ambientali, Laboratorio di Palinologia e Paleoecologia, Piazza della Scienza 1, 20126 Milano, Italy, [52]Department of Earth Sciences, Palynology and Palaeobotany Section, National Museums of Kenya, P.O. Box



40658, Nairobi, 00100, Kenya, [53]Department of Plant Sciences, University of the Free State, P.O Box
339, Bloemfontein, South Africa, [54]Graduate School of Life and Environmental Sciences, Kyoto
Prefectural University, 1-5 Hangi-cho, Shimogamo, Sakyo-ku, Kyoto 606-8522, Japan, [55]Department
of Geography, University of Exeter, Amory Building, Rennes Drive, Exeter, EX4 4RJ, UK, [56]Department
of Paleoecology and Landscape Ecology, Institute for Biodiversity and Ecosystem Dynamics,
Universiteit van Amsterdam, Science Park 904, 1098 XH Amsterdam, The Netherlands, [57]Department
of Biological Sciences, Florida Institute of Technology, Melbourne, FL, USA, [58]GNS Science1 Fairway
Drive, Avalon PO Box 30-368, Lower Hutt, New Zealand [59]Department of Earth Sciences, Montana
State University, Bozeman, MT 59717, USA, [60]U.S. Geological Survey, 926A National Center, Reston,
VA 20192, USA.

*Correspondence to*: Maria F. Sanchez Goñi (mf.sanchezgoni@epoc.u-bordeaux1.fr)

**Abstract**
Quaternary records provide an opportunity to examine the nature of the vegetation and fire
responses to rapid past climate changes comparable in velocity and magnitude to those expected in
the 21$^{st}$ century. The best documented examples of rapid climate change in the past are the warming
events associated with the Dansgaard-Oeschger (D-O) cycles during the last glacial period, which
were sufficiently large to have had a potential feedback through changes in albedo and greenhouse
gas emissions on climate. Previous reconstructions of vegetation and fire changes during the D-O
cycles used independently constructed age models, making it difficult to compare the changes
between different sites and regions. Here we present the ACER (Abrupt Climate Changes and
Environmental Responses) global database which includes 93 pollen records from the last glacial
period (73-15 ka) with a temporal resolution better than 1,000 years, 32 of which also provide
charcoal records. A harmonized and consistent chronology based on radiometric dating ($^{14}$C,
$^{234}$U/$^{230}$Th, OSL, $^{40}$Ar/$^{39}$Ar dated tephra layers) has been constructed for 86 of these records, although
in some cases additional information was derived using common control points based on event
stratigraphy. The ACER database compiles metadata including geospatial and dating information,
pollen and charcoal counts and pollen percentages of the characteristic biomes, and is archived in
*Microsoft Access*$^{TM}$ at https://doi.org/10.1594/PANGAEA.870867.




## 1. Introduction

There is considerable concern that the velocity of projected 21[st] century climate change is
too fast to allow terrestrial organisms to migrate to climatically suitable locations for their survival
(*Loarie et al., 2009; Burrows et al., 2011; Ordonez et al., 2013; Burrows et al., 2014*). The expected
magnitude and velocity of 21[st] century climate warming is comparable to abrupt climate changes
depicted in the geologic records, specifically the extremely rapid warming that occurred multiple
times during the last glacial period (Marine Isotope Stages 4 through 2, MIS 4-MIS2, 73,500–14,700
calendar years, 73.5–14.7 ka). The estimated increases in Greenland atmospheric temperature were
5–16°C [*Capron et al.*, 2010] and the duration of the warming events between 10 to 200 years
[*Steffensen et al.*, 2008]. These events are a component of longer-term millennial-scale climatic
variability, a pervasive feature through the Pleistocene [*Weirauch et al.*, 2008] which were originally
identified from Greenland ice archives [*Dansgaard et al.*, 1984] and in North Atlantic Ocean records
[*Bond and Lotti*, 1995; *Heinrich*, 1988] and termed Dansgaard-Oeschger (D-O) cycles and Heinrich
events (HE) respectively.
D-O events are registered worldwide, although the response to D-O warming events is
diverse and regionally specific (see e.g. [*Fletcher et al.*, 2010; *Harrison and Sanchez Goñi*, 2010;
*Sanchez Goñi et al.*, 2008]) and not a linear response to either the magnitude or the duration of the
climate change in Greenland. Given that the magnitude, length and regional expression of the
component phases of each of the D-O cycles varies [*Johnsen et al.*, 1992; *Sanchez Goñi et al.*, 2008],
they provide a suite of case studies that can be used to investigate the impact of abrupt climate
change on terrestrial ecosystems.
The ACER (Abrupt Climate change and Environmental Responses) project was launched in
2008 with the aim of creating a global database of pollen and charcoal records from the last glacial
(73 - 15 ka, kyr cal BP) which would allow us to reconstruct the regional vegetation and fire changes
in response to glacial millennial-scale variability, and evaluate the simulated regional climates



resulting from freshwater changes under glacial conditions. Although there are 232 pollen records
covering the last glacial period worldwide, only 93 have sufficient resolution and dating control to
show millennial-scale variability [*Harrison and Sanchez Goñi*, 2010].  It was necessary to re-evaluate
and harmonize the chronologies of these individual records to be able to compare patterns of change
from different regions. In this paper, we present the ACER pollen and charcoal database, including
the methodology used for chronological harmonization and explore the potential of this dataset by
comparing two harmonized pollen sequences with other palaeoclimatic records. Such a comparison
illustrates the novel opportunities for the spatial analyses of global climate events using this research
tool.
**2. Data and methods**
**2.1. Compilation of the records**

The ACER pollen and charcoal database includes records covering part or all of the last glacial

period and with a sampling resolution better than 1,000 years. These records were collected as raw
data, through direct contact with researchers or from the freely available European and African
Pollen Databases. Four records were digitized from publications using the Grapher$^{TM}$ 12 (Golden
Software, LLC) because the original data were either lost (Kalaloch: [*Heusser*, 1972] and Tagua Tagua
[*Heusser*, 1990]) or are not publicly available (Lac du Bouchet [*Reille et al.*, 1998] and Les Echets [*de*
*Beaulieu and Reille*, 1984]).  These digitized records are available as pollen percentages rather than
raw counts. All the records are listed and described in Table S1 (supplementary material).

**2.2. Harmonization of database chronologies**

The chronology of each of the records was originally built as a separate entity. In order to

produce harmonized chronologies for the ACER database, decisions had to be made about the types
of dates to use, the reference age for modern, the choice of calibration curve, the treatment of
radiocarbon age reservoirs, and the software used for age-model construction.



Radiometric ages ($^{14}$C, $^{235}$U/$^{230}$Th, OSL, $^{40}$Ar/$^{39}$Ar) and radiometrically-dated tephras are

given preference in the construction of the age models. The tephra ages were obtained either
through direct $^{40}$Ar/$^{39}$Ar dating of the tephra or $^{14}$C dating of adjacent organic material (Table 1).
When a radiometric or tephra date was obtained on a unit of sediment, the depth of the mid-point of
this unit was used for the date in the age modelling. Both the age estimate and the associated errors
(standard deviation) are required for age-model construction. When the positive and negative
standard deviations were different, the larger value was used for age-model construction. In cases
where the error measurements on the radiometric dates were unknown (e.g. site F2-92-P29), no
attempt was made to construct a harmonized age model.

Measured $^{14}$C ages were transformed to calendar ages, to account for the variations in the

atmospheric $^{14}$C/$^{12}$C ratio through time. Radiocarbon ages from marine sequences were corrected
before calibration to account for the reservoir effect whereby dates have old ages because of the
delay in exchange rates between atmospheric $CO_2$ and ocean bicarbonate and the mixing of young
surface waters with upwelled old deep waters. We used the IntCal13 and Marine13 calibration
curves for terrestrial and marine $^{14}$C dates, respectively [*Reimer et al.*, 2013], which are the
calibration curves approved by the radiocarbon community [*Hajdas*, 2014]. Although studies have
shown that the radiocarbon ages of tree rings from the Southern Hemisphere (SH) are ca 40 yr older
than Northern Hemisphere (NH) trees formed at the same time [*Hogg et al.*, 2013], this difference is
smaller than the laboratory errors on most of the $^{14}$C dates and, since the Marine13 calibration curve
does not distinguish between SH and NH sites, we use the NH IntCal13 calibration curve for all the
records.

The Marine13 calibration curve includes a default 400 yr reservoir correction. We adjusted

this correction factor for all the twenty six marine records included in the database using the regional
marine reservoir age (ΔR) in the Marine Reservoir Correction Database
(http://calib.qub.ac.uk/marine/). For twenty marine records, the correction factor was based on a
maximum of the 20 closest sites within 1,000 km to a specific site; for the remaining 6 marine



records this factor was based on a maximum of the 20 closest sites within 3,000 km. When ΔRs were
homogeneous, a value ± 100 years, over this area we used the mean of the 10 sites within 100 km to
provide a reservoir correction for the site. When there was heterogeneity in ΔR values within the
3,000 km target area, we selected only the sites with homogeneous ΔR within 100-200 km. Temporal
variations of ΔR were not taken into account since they are currently not well established for many
locations.
For periods beyond the limit of $^{14}$C dating (~45 ka) and for the few records without
radiometric dating, additional chronological control points were obtained based on "event
stratigraphy", specifically the identification of D-O warming events and Marine Isotope Stage (MIS)
boundaries (Table 1). No assumption was made that core tops were modern for both marine and
terrestrial cores. The ages of D-O warming events and those of the MIS boundaries were based on
the stratigraphy of core MD95-2042, southern Iberian margin (Table 1). The similarity of the
planktonic foraminifera $\delta^{18}$O record from MD95-2042 to the $\delta^{18}$O record from Greenland allowed to
match ages of individual D-O cycles, while the benthic foraminifera $\delta^{18}$O record from MD95-204
allowed to match ages of MIS boundaries [*Shackleton et al.*, 2000]. Both D-O and MIS ages were
directly transferred to the MD95-2042 pollen record. The chronology of this pollen record was in turn
transferred to the other European pollen records assuming synchronous afforestation during D-O
warming. The uncertainties for the event-based ages up to D-O 17 are from data summarized in
*Wolff et al.* [2010] and from AICC_2012 in NGRIP ice standard deviation [*Bazin et al.*, 2013] for older
events.
Non-radiocarbon dates are presented in the same BP notation as radiocarbon
determinations. The modern reference date is taken as 1950 AD, since this is the reference date for
the GICC05 chronology [*Wolff et al.*, 2010]).
Bayesian age modeling (e.g. using OxCAL, Bchron or BACON) requires information about
accumulation rates and other informative user-defined priors [*Blaauw and Christen*, 2011] that is
difficult to obtain for the relatively long ACER records. Moreover, BACON and Bchron [*Haslett and*



*Parnell,* 2008*, Parnell et al.,* 2008] do not handle sudden shifts in accumulation rate very well, and
such shifts are not uncommon across deglaciation and stadial time periods. We therefore use the
classical age-modeling approach in the CLAM software [*Blaauw*, 2010], implemented in R (R version
3.3.1) [*R Development Core Team*, 2016], to construct the age model.
Several age models were built for each record using the calibrated distribution of the
radiometric dates: a) linear interpolation between dated levels; b) linear or higher order polynomial
regression; and c) cubic, smoothed or locally weighted splines (Table S2). Linear interpolation is
generally the most parsimonious solution for records with no age reversals. However, if any of the
regression or spline models provided a better fit to the calibrated age range of outliers from a linear
model, we selected the model that included most of the outliers. If none of the regression or spline
model provided a better fit, we used linear interpolation after excluding the outliers. The database
includes information on the single 'best' age-model and the 95% confidence interval estimated from
the 10,000 iterated age-depth models (weighted mean) for every sample depth.

**2.3 The Structure of the Database**
The ACER pollen and charcoal data set is archived in a *Microsoft Access*[TM] relational database.
There are six main tables (Fig. 1).



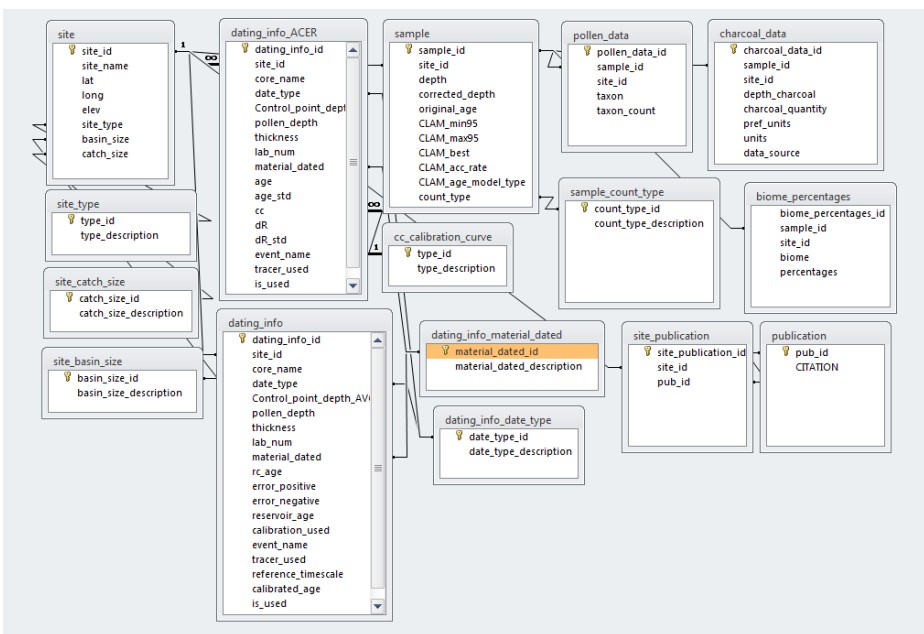


*Figure 1 – ACER database structure in ACCESS format.*

(1) Site Metadata. This table includes the original site name, geographical coordinates (latitude

and longitude in decimal degrees, elevation in meters above or below sea level) and additional meta-
data including site type (marine or terrestrial), basin size, catchment size. Basin size and catchment
size determine the size of the area sampled by the record (or pollen source area: see Prentice, 1988),
but are not always recorded in the original publication or known very accurately. A categorical
classification (small, medium, large, very large) is recorded in the database where these categories
are specified by ranges in km$^2$. The details of the original publication of the data are also given in this
table.

(2) Sample data.  The table records the identification number of each sample (sample id) at each

site (site id) and provides the depth of the sample (in cm from the surface). In only one site, core
MD04-2845, a corrected depth is provided on which the new age model is based. The pollen count
type (raw pollen count, pollen percentages given by the authors, or digitized percentage) is also
given. The original age of the sample according to the published age model when available and the



age determined from the best CLAM model (the min and the max at 95%, the accumulation rate and
the type of model used to obtain this age) are given.
(3) Pollen data. The pollen data are recorded as raw counts or as the pollen percentage of each
pollen and spore morphotype identified. The table records the identification number of each sample
(sample id), the taxon name and count/percentage. Although the taxon names were standardized
with respect to the use of terms such as type and to remove obvious spelling mistakes, no attempt
was made to ensure that the names are taxonomically correct.
(4) Charcoal data. The table records the identification number of each sample (sample id). The
charcoal data are recorded by depth (in cm from the surface), and information is given on the
quantity and unit of measurement, and data source. Charcoal abundance is quantified using a
number of different metrics, given for the majority in concentrations and for few of them in
percentages.
(5) Original dating information. This table contains information on dating for each core at each
site. The core name from the original publication is given, and the table provides information on date
type (conventional $^{14}$C, AMS $^{14}$C, $^{234}$U/$^{230}$Th, OSL, $^{40}$Ar/$^{39}$Ar, annual laminations, event stratigraphy,
TL), the average depth assigned to the data in the age-model construction, the dating sample
thickness, laboratory identification number, material dated (bulk, charcoal, foraminifera, pollen,
tephra, wood), measured radiometric age and associated errors.  The marine reservoir age (and
associated error) and the radiocarbon calibration curve used in the construction of the original age
model, and the original calibrated age, are also given. Dates that are based on recognized events are
also listed, and identified by the name of the event (event name) and the type of record in which it is
detected (tracer used). The column "is_used" corresponds to the dates used by the authors for
building the original age models.
(6) ACER dating information. The ACER dating information table duplicates the original dating
information file, except that it provides information about the explicit corrections and the
harmonized control points used to produce the ACER age models (Table 1). Specifically, it gives the



calibration curve used (no calibration, INTCAL13, MARINE13), and the local reservoir age (and
uncertainty) for marine cores.

*Table 1. Harmonized control points used for age models when radiometric ages ($^{14}$C, OSL, $^{40}$Ar/$^{39}$Ar,*
*$^{234}$U/$^{230}$Th) were not available.*

| Event stratigraphy[1,2,3,4,5,6] | | GICC05[8] b1950 Age ka | Tephra layers[8-19] | ACER chronology Age $^{14}$C[a] | ACER Age ka | Uncertainties[8,24] Years |
|---|---|---|---|---|---|---|
| | | | K-Ah[9] | 6.28 | | 130 |
| | | | Mazama Ash[10] | 6.84 | | 50 |
| | | | Rotoma[11] | 8.53 | | 10 |
| | | | U-Oki[12] | | 10[b] | 300 |
| Onset Holocene | | 11.65 | | | 11.65 | 50 |
| | | | Rotorua[11] | 13.08 | | 50 |
| MIS 1/2 | D-O 1 | 14.6 | | | 14.6 | 93 |
| | | | Rerewhakaaitu[13] | 14.7 | | 95 |
| | | | NYT[14] | | 14.9[b] | 400 |
| | | | Sakate[15] | 16.74 | | 160 |
| | | | Y-2[16] | 18.88 | | 230 |
| LGM | | | | | 21 | |
| | | | Kawakawa/Oruanui[17] | 21.30 | | 120 |
| | D-O 2 | 23.29 | | | 23.29 | 298 |
| MIS 2/3 | D-O 3 | 27.73 | | | 27.73 | 416 |
| | | | AT[9] | 24.83 | | 90 |
| | D-O 4 | 28.85 | | | 28.85 | 449 |
| | | | TM-15 | | 31[b22] | 8000 |
| | D-O 5 | 32.45 | | | 32.45 | 566 |
| | D-O 6 | 33.69 | | | 33.69 | 606 |
| | D-O 7 | 35.43 | | | 35.43 | 661 |
| | | | TM-18 | | 37[b22] | 3000 |
| | D-O 8 | 38.17 | | | 38.17 | 725 |
| | | | Y-5[16] | | 39.28[b] | 110 |
| | | | Akasuko[18] | 40.73 | | 1096 |
| | D-O 9 | 40.11 | | | 40.11 | 790 |
| | D-O 10 | 41.41 | | | 41.41 | 817 |
| | D-O 11 | 43.29 | | | 43.29 | 868 |
| | | | Breccia zone[18] | 43.29 | | 955 |
| | D-O 12 | 46.81 | | | 46.81 | 956 |
| | D-O 13 | 49.23 | | | 49.23 | 1015 |
| | D-O 14 | 54.17 | | | 54.17 | 1150 |
| | | | TM-19 | | 55[b22] | 2000 |
| | D-O 15 | 55.75 | | | 55.75 | 1196 |
| | D-O 16 | 58.23 | | | 58.23 | 1256 |
| MIS 3/4 | D-O 17 | 59.39 | | | 59.39 | 1287 |

| | | | | | |
|---|---|---|---|---|---|
| | onset HS 6 | 64.6[6] | | 64.6 | *1479* |
| | D-O 18 | 65[6] | | 65 | *1518* |
| MIS 4/5 | D-O 19 (onset Ognon II) | 72.28 | | 72.28 | *1478* |
| | D-O 20 (onset Ognon I) | 76.4 | | 76.4 | *1449* |
| | C 20 (stadial I) | 77[6] | | 77 | *1476* |
| | MS-insolation 15°S* | 81 | | 81 | *1504* |
| MIS 5.1 | D-O 21 (onset St Germain II) | 82.9[5] | | 82.9 | *1458* |
| | C 21 | 85[7] | | 85 | *1448* |
| | | | Vico[19] | 87[b] | 7000 |
| | | | Aso-4[20] | 89[b] | 7000 |
| | | | | | |
| | | | Ash-10[21] | 100[b] | 1540 |
| MIS 5/6 | | | | 135[23] | *2500* |


*Middle of "high" magnetic susceptibility record zone (consistently <50 SI units) tied to low in insolation for
January 15°S [*Gosling et al.*, 2008].
[a] Ages in $^{14}$C that were calibrated for the construction of the age model.
[b] Ages in $^{40}$Ar/$^{39}$Ar or $^{40}$K/$^{40}$Ar
K-Ah: Kikai-Akahoya; U-Oki: Ulleungdo-U4; NYT: Neapolitan Yellow Tuff ; AT: Aira Tephra; K-Tz: Kikai-
Tozurahara
[1][*Shackleton et al.*, 2000], [2][*Shackleton et al.*, 2004], [3][*Svensson et al.*, 2006], [4][*Svensson et al.*, 2008], [5][*Sánchez
Goñi*, 2007], [6][*Sanchez Goñi et al.*, 2013], [7][*McManus et al.*, 1994], [8][*Wolff et al.*, 2010], [9][*Smith et al.*, 2013], [10]
[*Grigg and Whitlock*, 1998], [11][*Newnham et al.*, 2003], [12][*Smith et al.*, 2011], [13][*Shane et al.*, 2003]; [14][*Deino et
al.*, 2004], [15][*Katoh et al.*, 2007], [16][*Margari et al.*, 2009]; [17][*Vandergoes et al.*, 2013]; [18][*Sawada et al.*, 1992],
[19][*Magri and Sadori*, 1999],[20][*Nakagawa et al.*, 2012], [21][*Whitlock et al.*, 2000], [22][*Wulf et al.*, 2004],;
[23][*Henderson and Slowey*, 2000], [24][*Bazin et al.*, 2013] (italics: uncertainties of the closest age in AICC_2012 in
NGRIP ice standard deviation).

Additional tables document the codes used in the main tables for e.g. basin type, basin size, date
type, material dated, calibration curve and biome percentage table that includes selected biomes
provided by the authors (Table 1). The taxa defining the pollen percentages of the main forest
biomes are those originally published by the authors in the Quaternary Science Reviews special issue
[*Fletcher et al.*, 2010; *Hessler et al.*, 2010; *Jimenez-Moreno et al.*, 2010; *Takahara et al.*, 2010]. The
taxa defining the pollen percentages of the main biomes from Africa (Mfabeni, Rumuiku) Australia
(Caledonia Fen, Wangoom) and New Zealand (Kohuora) not included in this issue are described in the
supplementary information.





Each table of the ACCESS database is also available as .csv file: a) Site, b) Sample (original depth-
age model and ACER depth-age model), c) Dating info (original dating information), d) dating info
ACER (harmonized dating information from this work), e) pollen data (raw data or digitized pollen
percentages; pollen percentages of different biomes) (Table 2), f) unique taxa in database (list of all
the identified taxa), g) charcoal data (raw or digitized).

*Table 2 – Biomes for which the pollen percentages data are included in the ACER database. Bo forest:*
*Boreal forest; Te mountain forest: Temperate mountain forest; Te forest: Temperate forest; WTe*
*forest: Warm-Temperate forest; Tr forest: Tropical forest; Subtr forest: Subtropical forest; SE Pine*
*forest: Southeastern Pine forest; Gr: Grasslands; Sav: Savanah. In Europe, Te forest includes*
*Mediterranean and Atlantic forests.*

| Europe | North America | Tropics | | East Asia | New Zealand | Australia |
| --- | --- | --- | --- | --- | --- | --- |
| | | American | African | | | |
| Te forest | Bo forest | Te mountain forest | | Bo forest | Te forest | WTe forest |
| | Te forest | WTe forest | | Te forest | | Te mountain forest |
| | WTe forest | Tr forest | | WTe forest | WTe forest | Sav |
| | SE Pine Forest | Gr | | Subtr forest | | |
| | | | | Gr | | |



**3 Results**
**3.1 The ACER pollen and charcoal database**
ACER database comprises all available pollen and charcoal records covering all or part of the
last glacial (73 to 15 ka) as of July 2015. It contains 93 well-resolved pollen records (< 1,000 years



between samples), 32 of which include charcoal data, from all the major potential present-day
biomes (Fig. 2). There are 2486 unique pollen and spore taxa in the database.

Harmonized age models were constructed for 86 out of the 93 records (Table S2 in the

supplementary information). The seven sites without harmonized age models are: F2-92-P29 (no
radiocarbon age errors available); Bear Lake (pollen was counted on one core but sample depths
could not be correlated with the cores used for dating); EW-9504 and ODP 1234 (original age models
based on correlation with another core, but tie point information was not available); Okarito Pakihi
(no dating information available) and Wonderkrater borehole 3 (multiple age reversals). The well-
known site of La Grande Pile [*de Beaulieu and Reille*, 1992] is not included in the ACER database
because the high-resolution data are not publicly available. Other sequences, such as Sokli in Finland,
were fragmented and could not be used (Table S1). These sites are shown at the bottom of the
supplementary Table S1.

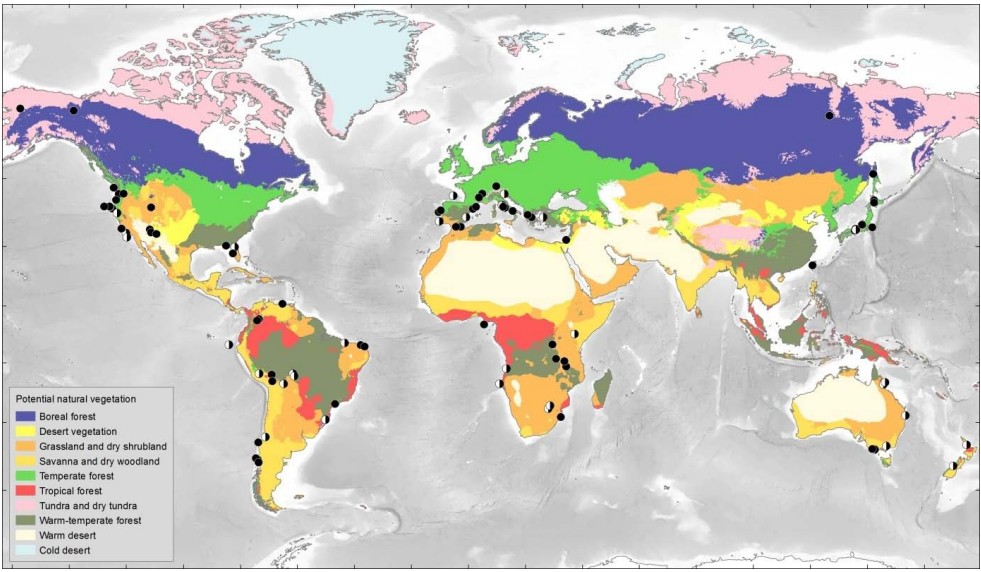


*Figure 2* - Map with location of the 93 marine and terrestrial sites (pollen: black circles, charcoal:
white circles) having resolution higher than 1 sample per 1000 years covering part or all the last
glacial (MIS 4, 3 and 2). Present-day potential natural vegetation after [*Levavasseur et al.*, 2012].




**3.2 Harmonized *versus* original age models**

We generated a total of 774 different age models. The age models of 45 records are based on
linear interpolation (Table S2 in the supplementary information). The age models of the other
records are derived from smooth or locally weighted splines (e.g. Lake Caço, Brazil; Fargher Lake,
North America; ODP1078C, southeastern Atlantic margin) or polynomial regression (e.g. Hanging
Lake and Carp Lake, North America; Lake Fuquene, Colombia; Valle di Castiglione, Europe) to include
as many as possible of the available radiometric dates. Since the focus for age modeling was the last
glacial period, age models for the Holocene (11.65ka - present) and Last Interglacial *sensu lato*
intervals (135ka -72.28 ka) are not necessarily well constrained.
Selected examples of the original and harmonized age models are illustrated in Figures 3 and
4. The original age model of marine core MD95-2043, western Mediterranean Sea (Figure 3a, red
curve) was based on tuning the mid-points of the cold to warm D-O transitions with the equivalent
mid-points in the alkenone-based sea-surface temperature (SST) record [*Cacho et al.*, 1999]. The
harmonized age model (black) is based on 21 $^{14}$C ages and two isotopic stratigraphic events (D-O 12
and D-O 14). The two age models are similar, with a mismatch of less than 1,000 years for periods
older than 35 ka and narrow uncertainties (Fig. 3a). In contrast, the original age model of the
terrestrial sequence of Valle di Castiglione, central Italy, published in Fletcher et al. (2010) differs
substantially, by several millennia, from the harmonized model in the interval between 50 and 30 ka
and has large uncertainties (Fig. 3b). This age model was based on two calibrated $^{14}$C dates, one
$^{40}$Ar/$^{39}$Ar tephra age (Neapolitan Yellow Tuff, Table 2) and the identification of D-O 8, 12 and 14 while
the new age model takes into account the entire number of $^{14}$C dates (eight), one $^{40}$Ar/$^{39}$Ar tephra
age and one GICC05-event stratigraphic age (identification of D-O 21). It derives from a 3$^{rd}$ order
polynomial regression model to take into account as many as possible of the radiometric ages
available (Table S2 in the supplementary information).




a.                                                                      b.

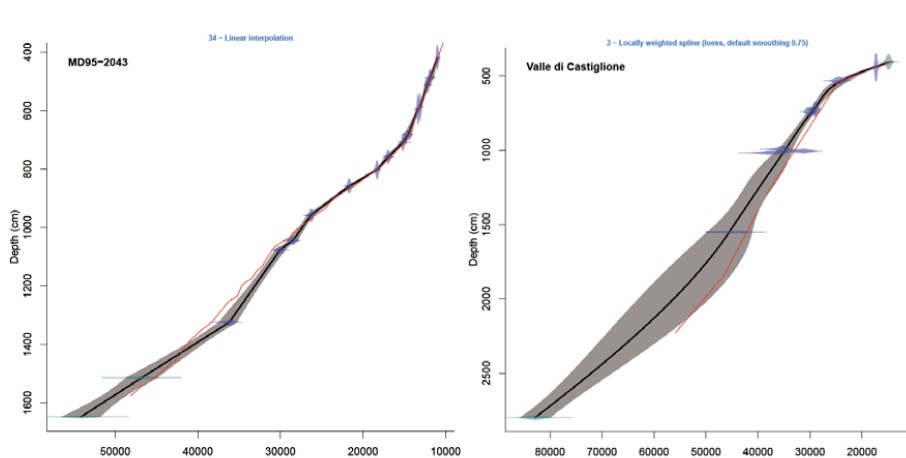


*Figure 3- a) Linear age model of the marine core MD95-2043, and b) 3<sup>rd</sup> order polynomial age model*

*of the terrestrial sequence Valle di Castiglione. Red line: original age model with the control points,*

*Black line: harmonized age model based on radiometric dating and event stratigraphy. Blue:*

*calibrated $^{14}$C distribution. Green: non-$^{14}$C age distribution ($^{40}$Ar/$^{39}$Ar, $^{234}$U/$^{230}$Th, OSL, event*

*stratigraphy). Grey shadow: age uncertainties.*


The original age model for marine core ODP 1233 C from the southern Pacific Ocean off

southern Chile was based on 19 AMS $^{14}$C dates calibrated using Calpal 2004 [*Heusser et al.*, 2006] and
is very similar to the harmonized age model (Figure 4a). The use of the new INTCAL13 calibration
curve is sufficient to explain the small differences between the original and harmonized age models.
In contrast, there are major differences between the original and harmonized age models for the
terrestrial pollen record of Toushe, Taiwan (Figure 4b). The original age model [*Liew et al.*, 2006] was
based on 24 uncalibrated radiometric dates for the 0-24 ka interval , and two dated isotopic events,
MIS 3/4 and MIS 4/5, which were dated following *Martinson et al.* [1987] to 58.96 ka and 73.91 ka
respectively. The harmonized age model is based on calibrated ages from 3 AMS $^{14}$C and 28



conventional $^{14}$C dates and dating of the MIS 3/4 and MIS 4/5 boundaries. In the ACER chronology,
these two events are dated to 59.39 ka and 72.28, respectively. In combination, these differences
produce substantially younger ages (by up to 5,000 years) for the interval between 50-26 ka than in
the original age model.







a.                                            b.

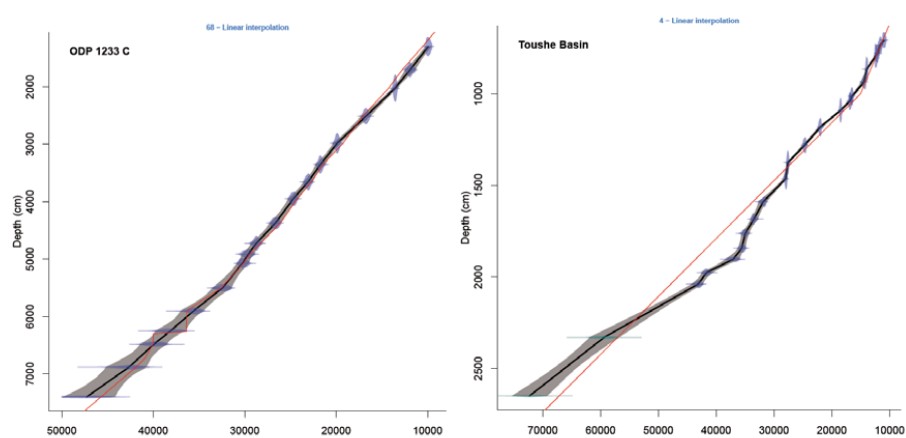


*Figure 4- a) Linear age model of the marine core ODP 1233 C, and b) Linear age model of the*
*terrestrial sequence Toushe (Taiwan). Red line: original age model with the control points, Black line:*
*harmonized age model based on radiometric dating. Blue: calibrated $^{14}$C distribution. Green: non-$^{14}$C*
*age distribution ($^{40}$Ar/$^{39}$Ar, $^{234}$U/$^{230}$Th, OSL, event stratigraphy). Grey shadow: age uncertainties.*



Figure 5 additionally illustrates pollen and microcharcoal data plotted against the harmonized age
models for few sites from different biomes. This figure highlights the regional response of the
vegetation and fire regime to the D-O events.



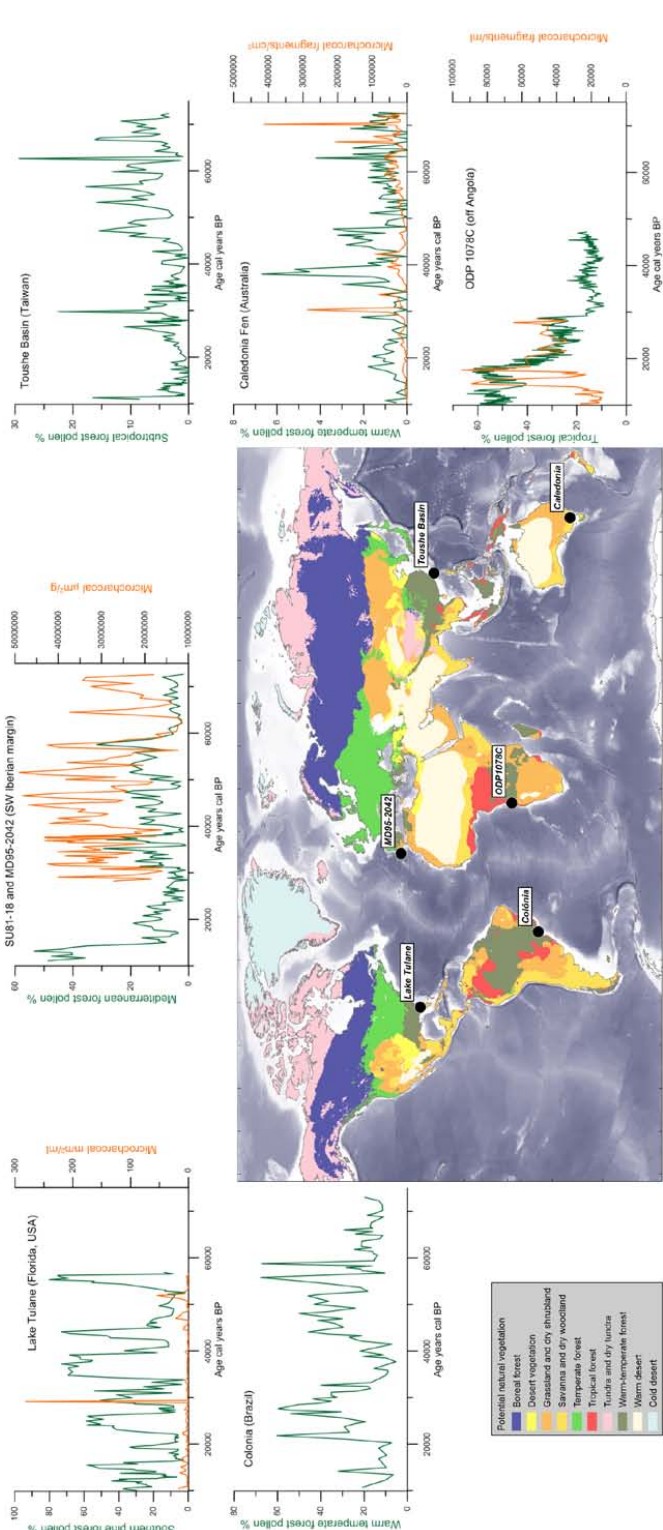




*Figure 5 – Pollen (black) and charcoal (orange) curves from six sites plotted against the harmonized*
*age model.*
**3.3 Vegetation and climate response to the contrasting D-O 8 and D-O 19 warming events.**
Comparison of the vegetation and climate response to warming events in two different
regions provides an example of the importance of developing harmonized chronologies. D-O 19 and
D-O 8 are iconic D-O events, characterized by strong warming in Greenland followed by long
temperate interstadials of 1,600 (GI 19) and 2,000 (GI 8) years respectively [*Wolff et al.*, 2010]. D-O 8
occurred ca 38.17 ka b1950 AD and was marked by an initial short-lived warming of ca 11°C, whereas
D-O 19 (ca 72.28 ka b1950 AD) was characterised by a maximum warming of ca 16°C. The difference
in the magnitude of warming suggests that the Northern Hemisphere monsoons would be stronger
during D-O 19 than D-O 8, but this is not consistent with speleothem evidence from Hulu Cave
(China) indicating that monsoon expansion was more marked during D-O 8 than during D-O 19
[*Wang et al.*, 2001] (Fig. 6). *Sanchez Goñi et al.* [2008] argued that the smaller increase in $CH_4$ during
D-O 19, by ca 100 ppbv, than during D-O 8, by ca 200 ppbv, was because the expansion of the East
Asian monsoon (and hence of regional wetlands) was weaker during D-O 19 due to the differences in
precession during the two events (Fig. 6). Differences in the strength of the monsoons between GI 8
(precession minima, high seasonality) and GI 19 (precession maxima, low seasonality) can also be
tested using evidence from the pollen record of Toushe Basin, which lies under the influence of the
East Asian monsoon. This record shows a similar development of moisture-demanding subtropical
forest, during the two interstadials (Fig. 6), and thus does not support the argument that the East
Asian monsoon was weaker/less expanded during GI 19 than during GI 8. However, Toushe Basin lies
in the tropical belt (23°N) and is likely to be less sensitive to changes in monsoon extent than more
marginal sites such as Hulu Cave (32°N).
Previous works have also hypothesized that the Mediterranean forest and climate were
tightly linked to the Asian and African monsoon through the Rodwell and Hoskins zonal mechanism



[*Marzin and Braconnot*, 2009; *Sanchez Goñi et al.*, 2008] or through shifts in the mean latitudinal
position of the ITCZ [*Tzedakis et al.*, 2009].  Data from Hulu cave [*Wang et al.*, 2001] and the western
Mediterranean region (MD95-2042 and SU81-18 twin pollen sequences) show that during warming
events occurring at minima in precession, such as D-O 8, monsoon intensification is stronger and
associated with a marked seasonality in the Mediterranean region (strong summer dryness) and,
therefore, a strong expansion of the Mediterranean forest and decrease in the summer dry-
intolerant Ericaceae (Fig. 6) [*Sánchez Goñi et al.*, 1999; *Sánchez Goñi et al.*, 2000]. Actually, we
observe parallel strong and weak increases in East Asian monsoon and Mediterranean forest during
GI 8 and GI 19, respectively. However, here again there is a discrepancy between the harmonized
Toushe pollen sequence and that from the Hulu cave and the western Mediterranean region: the
Mediterranean forest and monsoon during D-O 8 strongly increased while the subtropical forest
cover weakly expanded. The different latitudinal position of the Toushe Basin (23°N) in tropical
region and that of the Hulu Cave (32°N) and the southern Iberian margin sequence (37°N) both in the
subtropical region could explain such a discrepancy. A comprehensive analysis of differences in the
magnitude of monsoon expansion between D-O 8 and D-O 19 is now possible because of the creation
of robust and standardised age models for the ACER records.






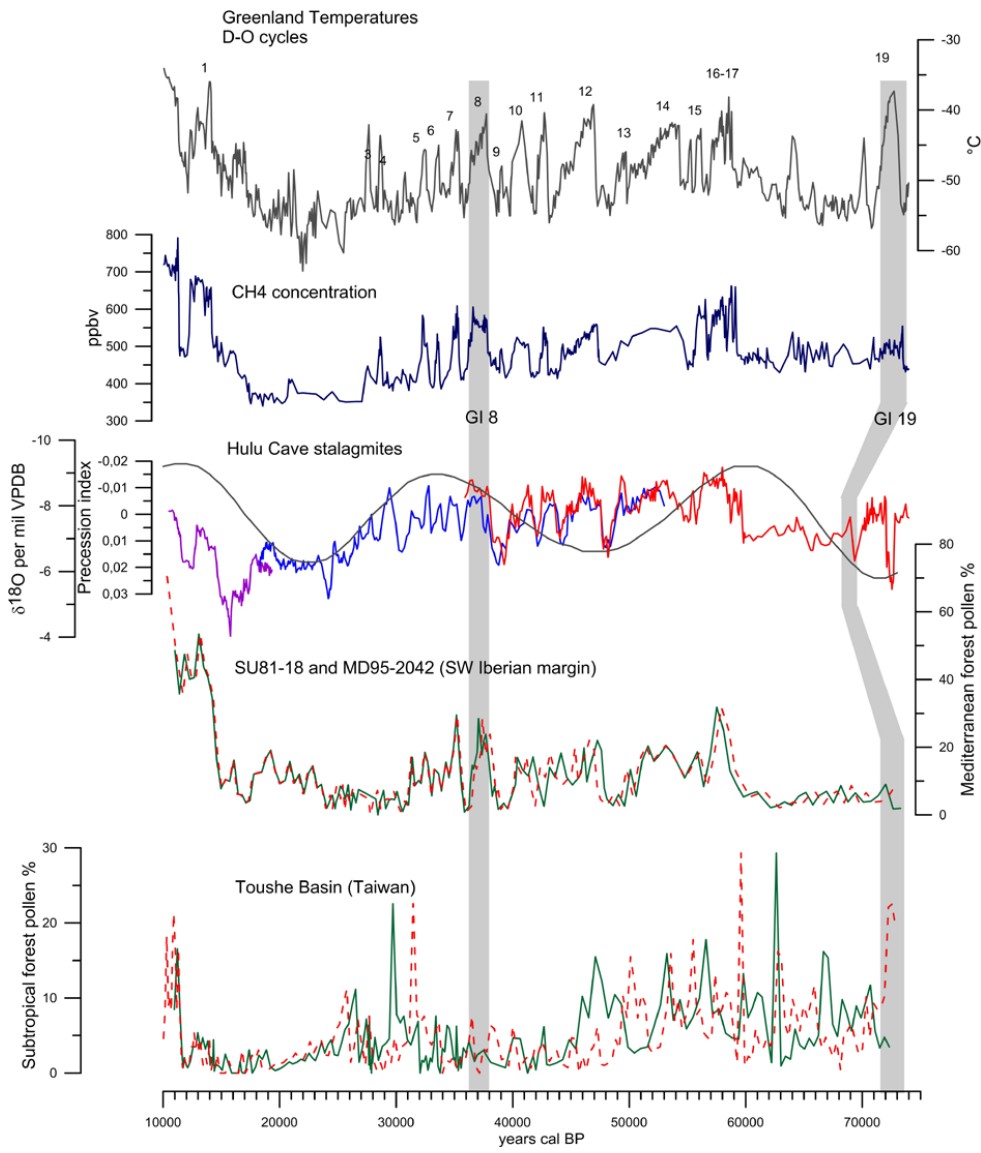

*Figure 6 - Comparison of pollen sequences from the Toushe Basin (Taiwan) and the SW Iberian margin*

*(cores MD95-2042 [Desprat et al., 2015; Sanchez Goñi et al., 2008] and SU 81-18 (23500-10000 cal*

*years BP) [Lézine and Denèfle, 1997]) for the interval 73-23.5 ka. Green line: new harmonized age*

*model, red dashed line: original age model. Grey vertical bands indicate the duration of GI 8, GI 16-17*

*and GI 19. Also shown the comparison with the Greenland temperature record (black) [Huber et al.,*

*2006; Landais et al., 2005; Sanchez Goñi et al., 2008], atmospheric CH$_4$ concentration (blue) record*





*[Chappellaz et al., 1997; Flückiger et al., 2004], compiled Hulu Cave $\delta^{18}O$ speleothem records (PD in*
*purple, MSD in green, and MSL in blue) [Wang et al., 2001], and precession index [Laskar et al.,*
*2004]. Note the mismatch in the timing of GI 19 between the Greenland and pollen harmonized age*
*models and the chronology of Hulu Cave.*
**4. Conclusions**
The ACER pollen and charcoal database (ACER 1.0) comprises all available pollen and charcoal
records covering part or all of the last glacial, as of July 2015. We foresee future updates of the ACER
database by the research community with newly published pollen and charcoal records. For
consistency age models for new sites should be constructed using the strategy described here.
The harmonization of the ACER age models in the ACER 1.0 database increases the
consistency between records by (a) calibrating all the radiocarbon dates using the recommended
INTCAL13 and MARINE13 calibration curves, (b) using the same ages for non-radiometric control
points and basing these on the most recent Greenland ice core chronology (GICC05), and (c) using
the CLAM software to build the age models and taking account of dating uncertainties. While these
harmonized age models may not be better than the original models, they have the great advantage
of ensuring comparability between pollen and charcoal records from different regions of the world.
As we have shown in the preliminary analyses of monsoon-related vegetation changes during D-O 8
and D-O 19, this will facilitate regional comparisons of the response to rapid climate changes.
The same strategy for age-model harmonization is now being applied to the sea-surface
temperature records from the last glacial that have been compiled by the ACER-INTIMATE group
(http://www.ephe-paleoclimat.com/acer/ACER%20INTIMATE.htm). This will ensure that the
terrestrial and marine databases share a common chronological framework, a considerable step
towards improving our knowledge of the interactions between oceans and land that underlie the
nature and timing of abrupt climatic changes.




**Data availability**
Supplementary data are available at https://doi.org/10.1594/PANGAEA.870867
**Author contributions.** MFSG, SD and ALD, developed the harmonized age models, ALD developed the
ACER database in ACCESS, FB participated in the construction of age models, JMPM extracted the
pollen percentage of the dominant biomes from the European sequences compiled in the ACER
database. MFSG and SPH write the manuscript. The remaining authors are listed alphabetically and
are data contributors (see their respective dataset on Table S1 in the Supplement link). All data
contributors (listed on Table S1) were contacted for authorisation of data publishing and offered co-
authorship. All the authors have critically reviewed the manuscript. Any use of trade, firm, or product
names is for descriptive purposes only and does not imply endorsement by the U.S. Government.


**Acknowledgements**
We wish to thank the QUEST-DESIRE (UK-France) bilateral project, the INQUA International Focus
Group ACER and the INTIMATE-COST action for funding a suite of workshops to compile the ACER
pollen and charcoal database and the workshop on ACER Chronology that allow setting the basis for
harmonizing the chronologies. We thank Maarten Blaauw for constructive discussions leading to the
construction of age models. JMPM was funded by a Basque Government post-doctoral fellowship
(POS_2015_1_0006) and SPH by the ERC Advanced Grant GC2.0 : Unlocking the past for a clearer
future. We thank V. Hanquiez for drawing Figure 2.




**Figures & Tables**
Figure 1 – ACER database structure in ACCESS format.

Figure 2 – Map with location of the 93 marine and terrestrial pollen sites covering part or all the last
glacial (MIS 4, 3 and 2). Sites have better resolution than 1 sample per 1000 years. Present-day
potential natural vegetation after [*Levavasseur et al.*, 2012].

Figure 3 –a) Linear age model of the marine core MD95-2043, and b) 3$^{rd}$ order polynomial age model
of the terrestrial sequence Valle di Castiglione (Italy). Red line: original age model with the control
points, Black line: harmonized age model with based on radiometric dating and event stratigraphy.
Blue: calibrated $^{14}$C distribution. Green: non-$^{14}$C age distribution (Ar/Ar, OSL, event stratigraphy).
Grey shadow: age uncertainties.

Figure 4- a) Linear age model of the marine core ODP 1233 C, and b) Linear age model of the
terrestrial sequence Toushe (Taiwan). Red line: original age model with the control points, Black line:
harmonized age model with based on radiometric dating and event stratigraphy. Blue: calibrated $^{14}$C
distribution. Green: non-$^{14}$C age distribution (Ar/Ar, OSL, event stratigraphy). Grey shadow: age
uncertainties.

Figure 5 – Pollen (black) and charcoal (orange) curves from six sites plotted against the harmonized
age model.

Figure 6 - Comparison of pollen sequences from the Toushe Basin (Taiwan) and the SW Iberian
margin (cores MD95-2042  [*Desprat et al.*, 2015; *Sanchez Goñi et al.*, 2008] and SU 81-18 (23500-
10000 cal years BP) [*Lézine and Denèfle*, 1997]) for the interval 73-23.5 ka . Green line: new
harmonized age model, red dashed line: original age model. Grey vertical bands indicate the duration



of GI 8, GI 16-17 and GI 19. Also shown the comparison with the Greenland temperature record
(black) [*Huber et al.*, 2006; *Landais et al.*, 2005; *Sanchez Goñi et al.*, 2008], atmospheric $CH_4$
concentration (blue) record [*Chappellaz et al.*, 1997; *Flückiger et al.*, 2004], compiled Hulu Cave $\delta^{18}O$
speleothem records (PD in purple, MSD in green, and MSL in blue)  [*Wang et al.*, 2001], and
precession index [*Laskar et al.*, 2004]. Note the mismatch in the timing of GI 19 between the
Greenland and pollen harmonized age models and the chronology of Hulu Cave.

Table 1. Harmonized control points used for age models when radiometric ages ($^{14}C$, OSL, $^{40}Ar/^{39}Ar$,
$^{234}U/^{230}Th$) were not available.

Table 2 – Biomes for which the pollen percentages data are included in the ACER database. Bo forest:
Boreal forest; Te mountain forest: Temperate mountain forest; Te forest: Temperate forest; WTe
forest: Warm-Temperate forest; Tr forest: Tropical forest; Subtr forest: Subtropical forest; SE Pine
forest: Southeastern Pine forest*;* Gr: Grasslands and dry shrublands; Sav: Savanah. In Europe, Te
forest refers to Mediterranean and Atlantic forests.





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
