# Peer review of "The ACER pollen and charcoal database: a global resource to document vegetation and fire response to abrupt climate changes during the last glacial period"

_Earth System Science Data, 2017_

## Referee Comment (RC1) · Anonymous Referee #1 · 23 Apr 2017

The authors present the development of a global database of pollen and charcoal records covering the past glacial period. The work is presented logically and there is a clear emphasis on the standardization of methods to allow consistent and comparable analysis with the database. The manuscript and results are clear and given the potentially broad uses of the database, the paper is ready for publication as it stands, barring a few minor corrections. My only general comment is that is it would have been useful to see some more detail on the potential uses of the database, but I understand that this is not the goal of the current paper.

[Figure]

Comments:

Line 139-140: "...opportunities for the spatial analyses of global climate events", should be changed to "spatial analyses of the impacts of global climate events"

Line 147: The criteria for sampling resolution of 1000 years is somewhat at odds with the goal stated in the abstract of being able to provide examples of climate change with similar rapidity to current changes. Some discussion of the ability of these records to resolve given events might be useful here

Line 202: While the assumption of synchronous afforestation is probably not an issue at these time scales (see previous comment), it is unlikely that the true afforestation was synchronous. Some discussion of the implications of this assumption in terms of interpreting the records would be useful

Line 222: How did the authors assess the quality of fit of the age model?

Section 2.3 Structure of the database. As I understand it, the database maps samples directly to site metadata. What happens if a site has multiple cores?

Line 250: What was the basis for standardizing taxa names? Within or across cores?

Line 297-298. I assume that the names in parentheses are site names, but please clarify this

Line 401 (figure 5 caption): I think it would be useful to change the color of the lines in the time plots to colors that are not used on the maps. As it stands, it looks like the green lines refer to the green biome. Also, please update the caption so the correct color is given for the pollen time series.

Line 439-441. A little more detail would be useful in understanding the potential of the database. How much further could you go? Are there other sites available?

Line 483. Change 'write' to 'wrote'

---

## Referee Comment (RC2) · T. Giesecke (Referee) · 5 Jul 2017

The manuscript describes a global collection of pollen and charcoal data for the last glacial, which is a welcomed and much needed research tool. The benefit lies in the collection of nearly all available continues records spanning the period, and the efforts of the lead authors to produce standard age models facilitating further analysis. In addition to radiometric ages, the lead authors also used climatic events as control points, which improve the individual chronologies; however, this prohibits the use of the database for few particular questions, which would require independent dating.

[Figure]

I have missed a short discussion on this problem, mentioning which climate events were mainly used and which type of analysis may be affected. Giesecke et al. (2014 VHA 23: 75-86) had to deal with the same problem and discuss some of the caveats. When trying to download the database I found it restricted. I assume this restriction will be lifted with the acceptance for publication, otherwise, you should specify how researchers might gain access to the database.

Specific and technical comments:

L. 167: Could uncertainties not be estimated from the type of age control available? See Giesecke et al. 2014 VHA

L. 199: Do you mean MD95-2042 as above mentioned?

Table 1: I find the uncertainty for the onset of the Holocene and some other events rather small could you comment on how they were derived or which uncertainty they represent?

L. 321: It is a pity that no age model could be produced for 7 out of 93 sites. Given the overall small number of sites, I wonder if the listed problems could not be overcome.

L. 326: Other sites co-authored by Reille were digitized why not this one? Moreover, one co-author of this manuscript was able to access the data (possibly by accident) and republished it (Helmens 2014 QSR), thus could not the percentage data be made available here?

L. 338: I don't understand why you generated 774 age models and find no explanation in the text.

L. 401: In my PDF the pollen curves appear green.

L. 408 and elsewhere: Why don't you use BP since you defined earlier that you use 1950 AD as zero.

---

## Author Comment (AC1) · 26 Jul 2017

Responses to the comments raised by the two referees

We thank both referees for their constructive comments, and address below the suggestions of change made by the two referees on a one to one basis.

Anonymous Referee #1

My only general comment is that is it would have been useful to see some more detail

on the potential uses of the database, but I understand that this is not the goal of the current paper.

Comments: Line 139-140: "... opportunities for the spatial analyses of global climate events", should be changed to "spatial analyses of the impacts of global climate events"

Done (L140-141).

Line 147: The criteria for sampling resolution of 1000 years is somewhat at odds with the goal stated in the abstract of being able to provide examples of climate change with similar rapidity to current changes. Some discussion of the ability of these records to resolve given events might be useful here.

We have modified the paragraph accordingly. Now it reads (L147-149): "The ACER pollen and charcoal database includes records covering part or all of the last glacial period and with a sampling resolution from few centuries to less than 1,000 years, which allows the identification of centennial-scale vegetation changes similar in duration to the current global climate change."

Line 202: While the assumption of synchronous afforestation is probably not an issue at these time scales (see previous comment), it is unlikely that the true afforestation was synchronous. Some discussion of the implications of this assumption in terms of interpreting the records would be useful.

The afforestation would be synchronous within the chronological uncertainty of the climatic events and the temporal resolution of the pollen records, i.e. few millennia at worst. We have reworded the paragraph that now reads (L203-211): "The chronology of this pollen record was in turn transferred to the other European pollen records assuming, within the uncertainties of the age models and the temporal resolution of the records, synchronous afforestation during D-O warming. Note that only a limited number of D-O events were used to constrain the age models of the European records (Table S1), i.e. all the afforestation episodes were not attributed to a D-O warming

event. The uncertainties for the event-based ages up to D-O 17 are from data summarized in Wolff et al. [2010] and from AICC_2012 in NGRIP ice standard deviation [Bazin et al., 2013] for older events. These uncertainties limit the discussion of the timing of European tree colonization during a D-O warming."

Line 222: How did the authors assess the quality of fit of the age model?

This information was not given in the submitted manuscript but it is included now in the revised version and new Table S2. In general, we have selected the age model that has the goodness-of-fit values among the lowest, but not necessarily the lowest. As stated in the submitted manuscript the linear interpolation was favored because it is generally the most parsimonious solution for records with no age reversals. For regression and spline models, the selection of the model was based on a combination of visual evaluation and goodness-of-fit values. For instance, in certain cases the model of lowest goodness-of-fit values presents unrealistic changes in sedimentation rates. The new paragraph reads as follows (L224-231): "For general guidance, we have selected the age model that has the goodness-of-fit values among the lowest, but not necessarily the lowest (Table S2). The linear interpolation model was favored because it is generally the most parsimonious solution for records with no age reversals or very abrupt age/depth change. For regression and spline models, the selection of the model was based on a combination of visual evaluation and goodness-of-fit values that allows the selection of the model including most of the outliers and realistic changes in sedimentation rates. When none of the regression or spline model provided a better fit, we excluded the age/depth outlying values proposed by the authors (Table S1). The database includes information on the single 'best' age-model and the 95% confidence interval estimated from the 10,000 iterated age-depth models (weighted mean) for every sample depth."

Section 2.3 Structure of the database. As I understand it, the database maps samples directly to site metadata. What happens if a site has multiple cores?

[Figure]

To clarify this issue we have added the following paragraph in the revised manuscript (L284-286): "The ACER database allows archiving multiple cores retrieved in the same site (site name followed by the name of the core). In such case, an age/depth model is constructed for each core and archived in the database." Lake Consuelo, Rice Lake and Wonderkrater have multiple cores (Consuelo: CON1 and CON2; Rice Lake: Rice Lake 79 and Rice Lake 81; Wonderkrater: borehole 3 and borehole 4) but composite depth records were lacking. Given the focus of this study, age models were created only for Lake Consuelo CON1 and Wonderkrater borehole 4 (CON2 and borehole 3 only cover the Holocene). Age models are available for the two cores from Rice Lake, with a hiatus between the two records of about 2000 years (oldest sample at 16000 in core 79 and youngest one at 18000 cal yr BP in core 81). These two cores have to be merged prior to future data analysis. We have added this information in the revised manuscript (L338-345).

Line 250: What was the basis for standardizing taxa names? Within or across cores?

The taxonomy was standardized across cores. We have modified the paragraph as follows (L259-262): "The taxa names were standardized across cores with respect to the accepted use of terms such as type presented in different pollen determination key publications (e.g. Moore et al. (1991), Faegri et al. (1989)), and abbreviations and obvious spelling mistakes were removed. The names in the database are listed in a new Excel file as supplementary information. "

Line 297-298. I assume that the names in parentheses are site names, but please clarify this.

Done (L310-311).

Line 401 (figure 5 caption): I think it would be useful to change the color of the lines in the time plots to colors that are not used on the maps. As it stands, it looks like the green lines refer to the green biome. Also, please update the caption so the correct color is given for the pollen time series.

We have drawn a new figure 5. In this figure, charcoal curves are in black and the color of pollen curves refers to the color of the biome type on the map (see also response to T. Giesecke). We have modified the legend of the figure accordingly. Now the legend reads: "Figure 5 – Pollen (green olive: warm-temperate forest; red: tropical forest) and charcoal (black) curves from six sites plotted against the harmonized age model."

Line 439-441. A little more detail would be useful in understanding the potential of the database. How much further could you go? Are there other sites available?

We have added a new paragraph that indicates other potentialities of the ACER database. This paragraph now reads (L465-471): "Besides the study of the monsoon variability, the ACER database has also the potential of reconstructing land-openess (land cover changes) trough time to evaluate changes in vegetation-albedo feedback, recovering the expansion and contraction of different taxa during the D-O cycles (i.e. using isolines), performing spatial climatic reconstructions and evaluating climate and vegetation models dealing with the rapid climatic variability. Construction of vegetation and climate maps for different time slices and evaluating model simulations can be continuously improved by implementing the ACER database version 1 (July 2015) with new chronologically harmonized sites."

Line 483. Change 'write' to 'wrote'

Done (L513).

T. Giesecke (Referee)

In addition to radiometric ages, the lead authors also used climatic events as control points, which improve the individual chronologies; however, this prohibits the use of the database for few particular questions, which would require independent dating. I have missed a short discussion on this problem, mentioning which climate events were mainly used and which type of analysis may be affected. Giesecke et al. (2014 VHA 23: 75-86) had to deal with the same problem and discuss some of the caveats. In the

revised version of the manuscript we have added a short discussion on that issue (see response to comment of Referee 1, Line 202).

When trying to download the database I found it restricted. I assume this restriction will be lifted with the acceptance for publication, otherwise, you should specify how researchers might gain access to the database.

The database will be freely available once the manuscript will be accepted.

Specific and technical comments:

L. 167: Could uncertainties not be estimated from the type of age control available? See Giesecke et al. 2014 VHA.

No, they cannot be estimated from the type of age control. Chronological points for site F2-92-P29 derived from 14C ages but the error measurements were not published and these data are lost.

L. 199: Do you mean MD95-2042 as above mentioned?

Yes

Table 1: I find the uncertainty for the onset of the Holocene and some other events rather small could you comment on how they were derived or which uncertainty they represent?

The small, less than 100 years, uncertainties are those published or given by the authors based on 14C, K/Ar and Ar/Ar age estimations, and ice layer counting of Greenland ice cores for the Holocene and D-O 1 (Table 1).

L. 321: It is a pity that no age model could be produced for 7 out of 93 sites. Given the overall small number of sites, I wonder if the listed problems could not be overcome.

These problems cannot be solved unfortunately, and we specify clearly in the submitted manuscript what kind of information lacks for building an age model.

L. 326: Other sites co-authored by Reille were digitized why not this one? Moreover, one co-author of this manuscript was able to access the data (possibly by accident) and republished it (Helmens 2014 QSR), thus could not the percentage data be made available here?

La Grande Pile pollen diagram could not be digitized due to the poor quality of the pdf support (added in L346-347). Helmens (2014) obtained the restricted data from the authors. At the time of the compilation of the ACER database these data were still restricted and we were not able to integrate them in the database. Now the data are in PANGEA. Hopefully, the version 2 of the ACER database will be implemented by including this site and other ones published since July 2015.

L. 338: I don't understand why you generated 774 age models and find no explanation in the text.

We generate 774 age models because we applied the different models available in CLAM (linear interpolation, linear regression, three orders of higher polynomial regression, cubic spline, and two smooth spline at 0.3 and 0.6 degree of freedom and locally weighted spline with different smoothing) to each of the 86 sites compiled. We have added this explanation in the revised manuscript (L358-361).

L. 401: In my PDF the pollen curves appear green.

Yes, there is a mistake. Following referee 1 comment we have changed the color of the curves. We have applied the biomes' color on the map, the curve of warm-temperate forest is now in green olive and that of tropical forest in red. The charcoal curves appear in black. We have modified the legend figure accordingly.

L. 408 and elsewhere: Why don't you use BP since you defined earlier that you use 1950 AD as zero.

Done (L431-432).

Please also note the supplement to this comment:
https://www.earth-syst-sci-data-discuss.net/essd-2017-4/essd-2017-4-AC1-supplement.pdf

[Figure]

**Fig. 1.** Figure 5 revised

**Supplement:**

**The ACER pollen and charcoal database: a global resource to document vegetation and fire response to abrupt climate changes during the last glacial period**

**ACER Project Members\*:** M.F. Sánchez Goñi[1,2], S. Desprat[1,2], A.-L. Daniau[3], F. Bassinot[4], J.M. Polanco-Martínez[2,5], S.P. Harrison[6,7], and ACER contributors

**Supplementary Information**

Taxa defining the pollen percentages of the main biomes in South Africa, Kenya, Australia and New Zealand not included in the QSR special issue (Sánchez Goñi and Harrison, 2010).

**Mfabeni Peatland (South Africa)**

Temperate savannah: Anacardiaceae, Ericaceae, Euphorbiaceae, Fabaceae, Fabaceae (*Acacia*), Proteaceae.

 Warm-temperate mixed forest:  Apocynaceae, Celastraceae, Combretaceae, Cyanthaceae, Erythroxylaceae, Flacourtiaceae, Moraceae, Myricaceae, Myrtaceae, Podocarpaceae, Rosaceae, Rubiaceae.

**Rumuiku Swamp (Kenya)**

Temperate forest : *Ilex, Celtis, Lannea*, Malvaceae, Rubiaceae, *Rhus, Rubus, Stoebe, Merrema*, Tiliaceae, *Oenostachys, Commelina, Abutilon, Clematis, Cissampelos, Cardamine*, Amaranthaceae/Chenopodiaceae, Acanthaceae, *Cleome, Cocculus, Plectranthus*, Cucurbitaceae, Caryophyllaceae, *Cuscuta, Kedrostis, Ranunculus, Gynandropsis*, Iridaceae, *Hygrophila, Heliotropium, Leucas*, Lamiaceae, Liliaceae, Fabaceae, *Trema, Valeriana, Ipomoea, Solanum*, Urticaceae, Ericaceae, Asteraceae, Brassicaceae, Apiaceae, *Artemisia*, Poaceae

Warm temperate forest: *Dombeya, Myrica, Nuxia, Olea*, Moraceae, *Podocarpus, Polyscias, Protea, Schefflera, Hagenia, Alchornea, Ilex, Macaranga, Afrocrania, Celtis, Croton, Juniperus*, Rubiaceae, *Rapanea, Lasianthus, Syzygium,* Capparidaceae, *Allophylus, Apodytes, Hypericum, Acalypha, Albizia, Antidesma, Acacia, Bosquea, Canthium, Cliffortia, Neoboutonia, Clausena, Combretum, Clerodendron, Cordia, Drypetes, Dracaena, Phyllanathus, Elatine, Ekebergia, Euclea, Faurea, Gunnera, Gnidia, Ziziphus, Lannea*, Malvaceae, *Maesa, Phyllanthus, Prunus, Ruelia*, Rutaceae, Rubiaceae, *Rhus, Rubus*, Sapindaceae, Sapotaceae, *Tapinanthus, Merremia*, Tiliaceae, *Oenostachys, Commelina, Abutilon, Clematis, Cissampelos, Cardamine*, Amaranthaceae/Chenopodiaceae, *Ricinus*, Acanthaceae, *Cleome, Cocculus, Plectranthus*, Cucurbitaceae, Caryophyllaceae, *Cuscuta, Chlorophytum, Corchorus, Kohautia, Vernonia, Pavetta, Anthospermum, Ranunculus, Galium, Gynandropsis*, Iridaceae, *Hyptis, Hygrophila, Leucas*, Lamiaceae, *Hypoestes*, Fabaceae, *Trema, Valeriana, Ipomoea, Indigofera, Solanum*, Urticaceae, Ericaceae , Asteraceae, Brassicaceae, Apiaceae, *Artemisia*, Poaceae.

Tropical forest: *Dombeya, Myrica, Nuxia, Olea*, Moraceae, *Podocarpus, Polyscias, Protea, Schefflera, Hagenia, Alchornea, Ilex, Macaranga, Afrocrania, Celtis, Croton, Juniperus*, Rubiaceae, *Rapanea, Lasianthus, Syzygium,* Capparidaceae, *Allophylus, Apodytes, Hypericum, Acalypha, Albizia, Antidesma, Acacia, Bosquea, Canthium, Cliffortia, Neoboutonia, Clausena, Combretum, Clerodendron, Cordia, Drypetes, Dracaena, Phyllanathus, Elatine, Ekebergia, Euclea, Faurea, Gunnera, Gnidia, Ziziphus, Lannea*, Malvaceae, *Maesa, Phyllanthus, Prunus, Ruelia*, Rutaceae, Rubiaceae, *Rhus, Rubus*, Sapindaceae, Sapotaceae, *Tapinanthus, Merremia*, Tiliaceae, *Oenostachys, Commelina, Abutilon, Clematis, Cissampelos, Cardamine*, Amaranthaceae/Chenopodiaceae, *Ricinus*, Acanthaceae, *Cleome, Cocculus, Plectranthus,* Cucurbitaceae, Caryophyllaceae, *Cuscuta, Chlorophytum, Corchorus, Kohautia, Vernonia, Pavetta, Anthospermum, Galium, Gynandropsis*, Iridaceae, *Hyptis, Hygrophila,* Lamiaceae, *Hypoestes*, Fabaceae, *Ipomoea, Indigofera,* Ericaceae, Asteraceae, Brassicaceae, Apiaceae, Poaceae.

**Caledonia Fen and Wagoom (Australia)**

Warm temperate forest: *Podocarpus*, *Phylloclades.*

Savannah: *Eucalyptus, Casuarina,* Poaceae, Asteraceae, Apiaceae, *Banksia, Pomaderris, Acacia, Dodonaea, Plantago*.

**Kohuora (New Zealand)**

Warm temperate forest: *Agathis, Alectryon, Ascarina, Dacrydium, Dacrycarpus*, Dodonaea, *Elaeocarpus,Griselinia, Knightia, Laurelia, Leucopogon fasciculatus, Libocedrus plumosa, Metrosideros, Metrosideros excelesa type, Neomyrtus, Nestegis, Phyllocladus trichomanoides, Plagianthus, Podocarpus, Prumnopitys taxifolia, Prumnopitys ferruginea, Pseudopanax, Weinmannia, Cyathea dealbata* type, *Cyathea smithii* type.

Temperate forest: *Fuscospora, Griselinia, Halocarpus bidwillii, Hoheria, Lagarostrobos, Lepidothamnus, Libocedrus bidwillii, Muehlenbeckia, Nothofagus menziesii, Phyllocladus alpinus, Plagianthus, Podocarpus, Quintinia.*

Table S1 – List of the applied and selected age models for the sites included in the ACER database.

LI: Liner interpolation; LR: Linear regression; PR2: Polynomial regression-order 2; PR3: Polynomial regression-order 3; PR4: Polynomial regression-order 4; CS: Cubic spline; SS0.3: Smooth spline (smoothing 0.3); SS0.6: Smooth spline (smoothing 0.6); LW0.75: Locally weighted spline (smoothing 0.75). Green cells indicate the selected age model.

No new age model for the following sites: Bear Lake; Lago Grande di Monticchio (too many major inversions in the 14C dates); Okarito Pakihi (lack of dating information); EW9504-17PC; F2-92-P29; ODP 1234; Wonderkrater (Borehole 3); Huiñamarca (Lake Titicaca, lacking dating uncertainties for tephra and U/Th dates).

| site_id | site_name | Linear interpolation | Linear regression | Polynomial regression-order 2 | Polynomial regression-order 3 | Polynomial regression-order 4 | Cubic spline | Smooth spline (default smoothing 0.3) | Smooth spline (smoothing 0.6) | Locally weighted spline (loess, default smoothing 0.75) | Outliers (depth cm) | Dating control | Comments |
|---|---|---|---|---|---|---|---|---|---|---|---|---|---|
| 1 | Abric Romaní | | | | x | x | | | x | **x** | | | linear interpolation was ok graphically but CLAM warned about too many age reversals |
| 79 | Azzano Decimo | **x** | | | | | | | | | 3233, 3342, 3464 | | 3 outliers |
| 6 | Bear Lake (BL00-1E) | | | | | | | | | | | | NO NEW AGE MODEL |
| 60 | Caço | | | x | | | | x | | | | | |
| 52 | Caledonia Fen | | | x | | | | | **x** | | 1676.5,304,305 | | 3 outliers |
| 81 | Cambara do Sul | **x** | | | | | | | | | | | |
| 41 | Camel Lake | **x** | | | | | | | | | | | |
| 42 | Carp Lake | | | x | | x | | | **x** | | 1400,1630 | | 2 outliers |
| 61 | Colônia | **x** | | | | | | | | | | | |
| 75 | Core Trident 163 31B | | | | | | x | x | | | | | |
| 58 | EW9504-17 PC | | | | | | | | | | | | NO NEW AGE MODEL |
| 59 | F2-92-P29 | | | | | | | | | | | | NO NEW AGE MODEL |
| 53 | F2-92-P3 | x | | | | | | **x** | | | | | |
| 7 | Fargher Lake | | | x | x | | | | **x** | | 866.5 | | 1 outlier |
| 66 | Fundo Nueva | **x** | | | | | | x | | | | | |
| 43 | Fuquene | | | | | **x** | | | | | | | |
| 8 | Füramoos | **x** | | | | | x | | | | | | |
| 67 | GeoB1023 | **x** | | | x | | | x | | x | | | |
| 44 | GeoB3104 | **x** | | | | x | | x | | x | | | |
| 45 | GeoB3910-2 | **x** | | | x | | | x | | x | | | |
| 54 | Hanging Lake | | | **x** | | | | | | | | | no original age model (no calibration at this time) |
| 82 | Hay Lake | **x** | | | | | | x | | | | | |
| 93 | Huiñamarca (Lake Titicaca) | | | | | | | | | | | | NO NEW AGE MODEL (lacking dating uncertainties for tephra and U/Th dates) |
| 92 | Ioannina | **x** | | | | | | | | | | | |
| 9 | Iwaya | **x** | | | | | | x | | | | | |
| 10 | Joe Lake | | | | | | | x | **x** | | | | |
| 74 | Kalaloch | | | | | | | **x** | | x | | | |
| 11 | Kamiyoshi Basin (KY01) | x | | | x | | | **x** | | | | | |
| 12 | Kashiru Bog | **x** | | | | | x | x | | | 86.25,162.25,333,815, 518.5 | | 5 outliers |
| 13 | Kenbuchi Basin | **x** | | | | | | x | | | | | |
| 14 | Khoe | **x** | | | | | | x | | | | | |
| 15 | Kohuora | **x** | | | | | | | | | 40, 100, 200, 300,950,444, 480, 871 | | 8 outliers |
| 16 | Kurota Lowland | | | | | | | **x** | | | | | |
| 17 | KW31 | | | | x | | | | **x** | | | | |
| 62 | La Laguna | **x** | | | | | | x | | | | | |
| 76 | Lac du Bouchet | | | | | | | | **x** | x | 1064 | | 1 outlier |
| 18 | Lagaccione | **x** | | | x | x | | | | x | 2000 | | 1 outlier |
| 37 | Lago Grande di Monticchio | | | | | | | | | | | | NO NEW AGE MODEL |
| 83 | Laguna Bella Vista | | **x** | | | | | | | | 148 | | 1 hiatus (135 cm), 1 outlier |
| 84 | Laguna Chaplin | **x** | | | | | | x | | | 285.5, 296.5 | | 2 outliers |
| 19 | Lake Banyoles | **x** | | | x | x | | | x | | | | |
| 94 | Lake Billyakh | **x** | | | | | | | | | 842 | | 1 outlier |
| 26 | Lake Biwa (BIW95-4) | | | | | x | | **x** | | | | | |
| 85 | Lake Consuelo | x | | | | | | **x** | | x | 790 | | 1 outlier |
| 20 | Lake Malawi | | | **x** | | | x | | | | 660 | | 1 outlier |
| 21 | Lake Masoko | | | | | | | **x** | | | | | |
| 86 | Lake Nojiri | **x** | | | x | | | x | | | | | |
| 23 | Lake Tanganyika | **x** | | | | x | | x | | x | | | |
| 24 | Lake Tulane | | | | | | | | **x** | | 1644, 1684,1707 | | outlier 1644, 1684,1707 |
| 51 | Lake Wangoom LW87 core | | **x** | x | | | | x | x | | | | |
| 25 | Lake Xinias | **x** | | | | | | x | | | | | |
| 73 | Les Echets G - DIGI | | | | | | | **x** | | | 350 | | outlier 350 |
| 27 | Little Lake | | | x | **x** | | | | | | 1813 | | outlier 1813 |
| 28 | Lynchs Crater | | | | **x** | x | | | | x | | | |
| 29 | MD01-2421 | | | | | x | | **x** | | | | | |
| 96 | MD02-2579 | | | | **x** | | | | | | 249 | | 1 outlier |
| 30 | MD03-2622 Cariaco Basin | | | | | | | **x** | | | | | |
| 31 | MD04-2845 | | | | | | | **x** | | x | | | |
| 64 | MD84-629 | **x** | | x | | | | x | | x | | | |
| 32 | MD95-2039 | **x** | | | x | | | x | | x | | | |
| 33 | MD95-2042 | | | | x | x | | **x** | | x | | | |
| 34 | MD95-2043 | **x** | | | | | | x | | | | | |
| 35 | MD99-2331 | | | | x | | | | | **x** | | | |
| 36 | Megali Limni | | | | x | | | x | **x** | | | | |
| 87 | Mfabeni Peatland | **x** | | | | | | x | | | | | |
| 88 | Nakafurano | **x** | | | | | | x | | | | | |
| 89 | Native Companion Lagoon | | | | | | | | **x** | | 162.3 | | 1 outlier |
| 46 | Navarrés | | | | | | | **x** | | | | | |
| 68 | ODP 1233 C | **x** | | | x | x | x | x | | x | | | |
| 69 | ODP 1234 | | | | | | | | | | | | NO NEW AGE MODEL |
| 95 | ODP 820 | **x** | | | | | | | | | | | |
| 48 | ODP site 976 | | | | | | | **x** | | | | | |
| 55 | ODP1019 | | | | | | | x | **x** | | | | |
| 38 | ODP1078C | | | | | | | | **x** | x | | | |
| 47 | ODP893A | | | **x** | | | | | x | x | | | |
| 50 | Okarito Pakihi | | | | | | | | | | | | NO NEW AGE MODEL (lack of dating information) |
| 80 | Pacucha | **x** | | | | | | x | | | 1124, 1182 | | 2 outliers |
| 39 | Potato Lake | **x** | | | | | | | | | | | |
| 90 | Rice Lake (Rice Lake 79) | **x** | | | | | | | | | | | |
| 77 | Rumuiku Swamp | **x** | | | | | | | | | 1465 | | 1 outlier |
| 49 | Siberia | **x** | | | | x | | x | | | | | no original age model (no calibration at this time) |
| 40 | Stracciacappa | **x** | | | | | | x | | | | | |
| 63 | Tagua Tagua - DIGI | **x** | | | x | | | | | | | | |
| 70 | Taiquemo | | | | | | | | **x** | | | | |
| 4 | Toushe Basin | **x** | | | | | | x | x | | | | |
| 71 | Tswang Crater | **x** | | | | | | x | | | | | |
| 56 | Tyrrendara Swamp | **x** | | | | | | | | | 303 | | 1 outlier |
| 3 | Valle di Castiglione | | | | | | | x | | **x** | | | |
| 57 | W8709-13 PC | | | | **x** | | | | x | | | | |
| 91 | W8709-8 PC | | | x | | | | | | **x** | | | |
| 2 | Walker Lake | **x** | | | x | x | | x | | | | | continuous sequence up to 30 ka (hiatuses after) |
| 72 | Wonderkrater (Borehole 3) | | | | | | | | | | | | NO NEW AGE MODEL |
| 97 | Wonderkrater (Borehole 4) | **x** | | | | | | x | | | 705 | | 1 outlier |
| 98 | Rice Lake (Rice Lake 81) | **x** | | | | | | x | | | | | |
| 99 | SU 81-18 | | | x | | | | | **x** | | 551, 691 | | 2 outliers |

Green cells indicate the selected age model.

Table S2 – Goodness-of-fit for the selected age models. Green cells indicate the selected age model. Inf: infinite; NA: no available

| | A | B | C | D | E | F | G | H | I | J |
|---|---|---|---|---|---|---|---|---|---|---|
| 1 | sitename | Fit -linear | Fit -reg.lin | Fit -pol.reg2 | Fit -pol.reg3 | Fit -pol.reg4 | Fit -cubic | -smooth.spl( | -smooth.spl( | Fit -loess |
| 2 | Abric Romani | NA | 9.7 | 9.7 | 9.7 | 9.7 | NA | NA | 9.7 | 9.7 |
| 3 | Azzano Decimo | 7.94 | Inf | NA | NA | NA | NA | NA | NA | NA |
| 4 | Caco | NA | 127.36 | 95.9 | 110.38 | 123.66 | *NA* | 75.1 | 146.24 | NA |
| 5 | Caledonia Fen | NA | 311.52 | 104.57 | 170.58 | NA | NA | 28.76 | 76.86 | NA |
| 6 | Cambara do Sul | 2 | 226.95 | NA | 2.02 | NA | 2.05 | 6.97 | 175.94 | 2.02 |
| 7 | Camel Lake | 5.03 | 136.84 | 124.25 | NA | NA | NA | NA | 75.29 | NA |
| 8 | Carp Lake | NA | 109.94 | 62.27 | 55.42 | 50.09 | NA | NA | 53.84 | NA |
| 9 | Colonia | 7.21 | 1002.47 | 437.39 | 113.01 | NA | 7.15 | 20.37 | 574.57 | 43.97 |
| 10 | Core Trident 163 31B | NA | 40.23 | 33.32 | 33.9 | 31.06 | NA | 15.82 | 29.03 | 31.28 |
| 11 | F2-92-P3 | NA | 25.03 | 19.94 | 19.54 | 20.83 | NA | 13.96 | 18.54 | 20.22 |
| 12 | Fargher Lake | NA | 264.77 | 79.58 | 123.18 | NA | NA | NA | 74.12 | NA |
| 13 | Fundo Nueva | 2.72 | 67.9 | 19.7 | 23.07 | NA | NA | 15.08 | 58.39 | NA |
| 14 | Fuquene | NA | 215.7 | 203.56 | 208.83 | 169.95 | NA | NA | 192.66 | NA |
| 15 | Furamoos | 6.99 | 14.99 | 7.25 | NA | NA | 7.28 | NA | NA | 6.94 |
| 16 | GeoB1023 | 3.89 | 44 | 44.22 | 18.68 | 5.72 | 3.97 | 8.59 | 43.54 | 5.84 |
| 17 | GeoB3104 | 5.67 | 35.78 | 32.12 | 40.89 | NA | 5.9 | 14.77 | 31.36 | 23.79 |
| 18 | GeoB3910-2 | 7.03 | 38.78 | 49.75 | 34.15 | 7 | 6.95 | 8 | 39 | 5.06 |
| 19 | Hanging Lake | NA | 233.16 | NA | 60.1 | NA | NA | NA | 58.15 | NA |
| 20 | Hay Lake | 1.81 | 19.88 | 16.43 | 21.75 | NA | NA | 15.04 | 19.39 | NA |
| 21 | Ioannina | 5.92 | 54.56 | 226.59 | NA | NA | NA | 73.89 | 83.3 | NA |
| 22 | Iwaya | 2.34 | 19.67 | 27.97 | 2.68 | NA | 2.65 | 6.3 | 19.24 | 2.37 |
| 23 | Joe Lake | NA | 1417.94 | NA | NA | NA | NA | NA | 709.3 | NA |
| 24 | Kalaloch - DIGI | NA | 115.39 | 58.58 | NA | NA | NA | NA | 18.08 | 19.44 |
| 25 | Kamiyoshi Basin (KY01) | 5.06 | 10.29 | 3.62 | 4.16 | NA | NA | 4.47 | 7.49 | NA |
| 26 | Kashiru Bog | 5.83 | 199.12 | NA | NA | NA | 6.68 | 31.33 | 159.02 | NA |
| 27 | Kenbuchi Basin | 2.24 | 84.98 | 43.67 | NA | NA | 2.68 | 9.44 | 57.41 | NA |
| 28 | Khoe | 2.45 | 159.64 | 214.92 | NA | NA | NA | 11.95 | 165.05 | NA |
| 29 | Kohuora | 6.05 | Inf | NA | 144.92 | NA | NA | NA | 309.92 | NA |
| 30 | Kurota Lowland | NA | 40.04 | 125.5 | 17.75 | NA | NA | 5.84 | 44.41 | NA |
| 31 | KW31 | NA | 110.16 | 89.64 | 140.93 | NA | NA | NA | 98.54 | NA |
| 32 | La Laguna | 3.63 | 320.79 | NA | 117.21 | NA | NA | 56.39 | 233.54 | NA |
| 33 | Lac du Bouchet - DIGI | NA | 24.51 | 24.14 | 25.16 | 23.04 | NA | NA | 19.09 | 17.46 |
| 34 | Lagaccione | 8.35 | 79.33 | 66.41 | 30.7 | 28.2 | NA | NA | 37.4 | 31.82 |
| 35 | Laguna Chaplin | 5.52 | 223.01 | NA | NA | NA | 5.69 | 17.71 | 171.54 | NA |
| 36 | Lake Banyoles | 5.55 | 5.6 | 5.61 | 5.53 | 5.53 | NA | NA | 5.55 | NA |
| 37 | Lake Billyakh | 6.64 | 1253.57 | 1073.72 | NA | NA | NA | NA | 616.92 | NA |
| 38 | Lake Biwa (BIW95-4) | NA | 55.06 | 37.86 | 19.08 | 20.07 | NA | 16.7 | 32.23 | 22.41 |
| 39 | Lake Consuelo | NA | 291.67 | 25.17 | 23.27 | 25.76 | NA | 15.88 | 42.83 | 23.48 |
| 40 | Lake Malawi | 7.87 | 43.24 | 16.84 | NA | NA | 8.48 | NA | NA | NA |
| 41 | Lake Masoko | NA | 70 | 60.07 | 58.08 | NA | NA | 31.42 | 53.02 | NA |
| 42 | Lake Nojiri | 3.98 | 515.46 | 168.25 | 12.43 | 12.81 | NA | 12.98 | 18.65 | NA |
| 43 | Lake Tanganyika | 4.08 | 40.83 | NA | 7.49 | 5.01 | 3.58 | 4.41 | 20.66 | 3.76 |
| 44 | Lake Tulane | NA | Inf | Inf | Inf | Inf | NA | NA | Inf | NA |
| 45 | Lake Wangoom LW87 core | NA | 12.89 | 12.74 | 5 | 7.53 | NA | 11.34 | 12.74 | NA |
| 46 | Lake Xinias | 4.8 | 82.42 | 17.86 | 5.54 | NA | 5.35 | 11.51 | 68.71 | 5.07 |
| 47 | Les Echets G - DIGI | NA | 70.38 | NA | NA | NA | NA | 16.61 | 45.19 | NA |
| 48 | Little Lake | NA | Inf | 291.3 | 294.58 | 291.33 | NA | NA | 269.11 | NA |
| 49 | Lynchs Crater | NA | 381.48 | 214.42 | 177.09 | 182.27 | NA | NA | 178.26 | 168.91 |
| 50 | MD01-2421 | NA | 318.93 | 612.84 | 359.84 | 69.38 | NA | 23.74 | 238.3 | NA |
| 51 | MD02-2579 | NA | 1364.41 | NA | 127.39 | NA | NA | NA | 191.41 | NA |
| 52 | MD03-2622 Cariaco Basin | NA | 11.82 | 11.84 | 11.78 | 11.79 | NA | 11.78 | 11.8 | 11.78 |
| 53 | MD04-2845 | NA | Inf | 52.49 | 28.35 | 29.47 | NA | NA | 22.24 | 26.94 |
| 54 | MD84-629 | 2.77 | 2.91 | 2.9 | 2.77 | NA | 2.77 | 2.82 | 2.91 | 2.77 |
| 55 | MD95-2039 | 10.4 | 10.38 | 10.4 | 10.4 | 10.4 | NA | 10.4 | 10.4 | 10.4 |
| 56 | MD95-2042 | NA | 520.87 | 103.03 | 49.83 | 53.79 | NA | NA | 49.08 | 52.89 |
| 57 | MD95-2043 | 12.73 | Inf | 234.84 | 82.02 | NA | NA | 14.63 | 81.69 | 55.53 |
| 58 | MD99-2331 | NA | 613.51 | 169.98 | 113.75 | 111.31 | NA | NA | NA | 114.26 |
| 59 | Megali Limni | NA | 12.24 | 10.74 | 11.65 | 11.74 | NA | 5.87 | 9.88 | NA |
| 60 | Mfabeni Peatland | 3.08 | 145.52 | 150.1 | NA | NA | NA | 59.33 | 138.83 | 2.92 |
| 61 | Nakafurano | 3.3 | 278.83 | 348.19 | NA | NA | NA | 20.9 | 278.61 | NA |
| 62 | Native Companion Lagoon | NA | 72.74 | 32.09 | 48.52 | NA | NA | NA | 24.87 | NA |
| 63 | Navarres | NA | 548.05 | 615.85 | NA | NA | NA | 47 | 323.64 | 151.78 |
| 64 | ODP 1233 C | 11.56 | 22.5 | 26.55 | 25.91 | 18.58 | NA | 12.59 | 17.4 | 18.45 |
| 65 | ODP 820 | 4.46 | 22.27 | NA | 5.66 | NA | NA | NA | 2.78 | NA |
| 66 | ODP site 976 | NA | 216.2 | 133.71 | 136.6 | 112.46 | NA | 48.06 | 115.98 | 112.16 |
| 67 | ODP1019 | 7.97 | 418.97 | 36.02 | 31.25 | 31.26 | NA | 26.66 | 31.78 | 32 |
| 68 | ODP1078C | NA | 957.42 | 345.83 | NA | NA | NA | NA | 237.79 | 262.05 |
| 69 | ODP893A | NA | 122.22 | 28.51 | 24.11 | 23.68 | NA | 16.86 | 22.07 | 24.27 |
| 70 | Pacucha | 4.91 | 135.24 | 234.58 | 86.2 | 156.76 | NA | 42.77 | 140.32 | NA |
| 71 | Potato Lake | 2.81 | 31.59 | 32.73 | NA | NA | NA | NA | 29.77 | NA |
| 72 | Rice Lake (Rice Lake 79) | 1.57 | 1.58 | NA | NA | NA | 1.58 | NA | NA | NA |
| 73 | Rice Lake (Rice Lake 81) | 3.74 | Inf | NA | NA | NA | NA | 83.89 | Inf | NA |
| 74 | Rumuiku Swamp | 5.49 | 440.75 | 275.57 | 128.09 | NA | 6.02 | 66.3 | 434.58 | NA |
| 75 | Siberia | 6.67 | 305.02 | 197.77 | 122.46 | 119.85 | NA | 134.57 | 146.06 | NA |
| 76 | Stracciacappa | 3.27 | 41.6 | 37.87 | 47.99 | 32.3 | NA | 25.56 | 33.46 | 37.96 |
| 77 | SU81-18 | NA | 21.32 | 20.24 | 20.82 | NA | NA | NA | 18.38 | NA |
| 78 | Tagua Tagua - DIGI | 5.18 | 19.39 | 9.82 | 8.92 | 8.02 | NA | NA | 8.03 | 7.74 |

| | A | B | C | D | E | F | G | H | I | J |
|---|---|---|---|---|---|---|---|---|---|---|
| 79 | Taiquemo | NA | 889.16 | 452.44 | 232.04 | 245.97 | NA | NA | 116.87 | 182.74 |
| 80 | Toushe Basin | 14.55 | 132.02 | 170.44 | 170.5 | 78.18 | NA | 26.45 | 46.22 | 43.61 |
| 81 | Tswaing Crater | 2.08 | 175.16 | 112.94 | NA | NA | NA | 42.41 | 157.45 | 2.13 |
| 82 | Tyrrendara Swamp | 2.46 | 2.56 | 2.34 | NA | NA | 2.35 | NA | NA | 2.5 |
| 83 | Valle di Castiglione | NA | 64.92 | 44.71 | 15.46 | NA | NA | 8.18 | 17.67 | 7.72 |
| 84 | Walker Lake | 6.39 | 25.91 | 19.25 | 9.97 | NA | NA | 8.93 | 19.94 | NA |
| 85 | W8709-13 PC | NA | 94 | 72.59 | 81.3 | 62.6 | NA | NA | 61.9 | NA |
| 86 | W8709-8 PC | NA | 410.82 | 381.61 | 277.11 | NA | NA | 48.82 | 178.73 | NA |
| 87 | Wonderkrater (borehole 4) | 13.97 | 513.41 | NA | NA | NA | 14.38 | 40.7 | 348.38 | NA |
| 88 | Laguna Bella Vista | NA | 2.88 | NA | NA | NA | NA | NA | NA | NA |